# Democratization of PV Micro-Generation System Monitoring Based on Narrowband-IoT

**DOI:** 10.3390/s22134966

**Published:** 2022-06-30

**Authors:** José Miguel Paredes-Parra, Raquel Jiménez-Segura, David Campos-Peñalver, Antonio Mateo-Aroca, Alfonso P. Ramallo-González, Angel Molina-García

**Affiliations:** 1Technological Center of Energy and Environment (CETENMA), 30353 Cartagena, Spain; jmparedes@cetenma.es (J.M.P.-P.); david.campos@cetenma.es (D.C.-P.); 2Departament of Automatics, Electrical Engineering and Electronic Technology, Universidad Politécnica de Cartagena, 30202 Cartagena, Spain; raquel.jimenez@edu.upct.es (R.J.-S.); antonio.mateo@upct.es (A.M.-A.); 3Department of Information and Communications Engineering, Computer Science Faculty, University of Murcia, 30100 Murcia, Spain; alfonsop.ramallo@um.es

**Keywords:** photovoltaic microgeneration, NB-IoT technology, monitoring network, Internet of Things

## Abstract

Power system configuration and performance are changing very quickly. Under the new paradigm of prosumers and energy communities, grids are increasingly influenced by microgeneration systems connected in both low and medium voltage. In addition, these facilities provide little or no information to distribution and/or transmission system operators, increasing power system management problems. Actually, information is a great asset to manage this new situation. The arrival of affordable and open Internet of Things (IoT) technologies is a remarkable opportunity to overcome these inconveniences allowing for the exchange of information about these plants. In this paper, we propose a monitoring solution applicable to photovoltaic self-consumption or any other microgeneration installation, covering the installations of the so-called ’prosumers’ and aiming to provide a tool for local self-consumption monitoring. A detailed description of the proposed system at the hardware level is provided, and extended information on the communication characteristics and data packets is also included. Results of different field test campaigns carried out in real PV self-consumption installations connected to the grid are described and analyzed. It can be affirmed that the proposed solution provides outstanding results in reliability and accuracy, being a popular solution for those who cannot afford professional monitoring platforms.

## 1. Introduction

In recent years, most countries have established ambitious renewable energy targets for their electricity generation to foster sustainable and low-emission initiatives. In line with the European and COP21 decarbonization environmental objectives [1], Spain has promoted new legislation providing support and an optimized framework for the Renewable Energy Sources (RES) integration, mainly wind power plants and solar photovoltaic (PV) installations [2]. These new regulations, together with a significant reduction in the cost of PV technology as a result of their own technological maturity, have addressed the enormous growth in the integration of such solutions. Subsequently, a relevant number of PV installations can be found at different voltage levels; mainly under the alternative cooperatively-owned organizations giving a substantially different model of energy distribution and provision [3]. In Europe, and more specifically in Spain, PV installations have been widely integrated by prosumers as feasible strategies to decrease power demand, fulfill emission agreements and accelerate self-consumption growth rates [4]. These policies can have relevant consequences on the design and operation of distribution networks to which most prosumers are connected [5]. Under this scenario, guaranteeing a safe and reliable RES integration is a remarkable challenge to be solved by power systems in the coming years [6]. Actually, high RES integration is included in the Strategic Energy Technology Plan (SET Plan) of the European Union—see Activity 4: Operation and diagnosis of Photovoltaic Plants, highlights the digitization of the energy sector as a fundamental and crucial element for the EU [7]. In parallel to the current developments of technology, digitization offers new opportunities for system operators and facility managers and maintainers by optimizing the operation of their assets, reducing their operating costs and allowing greater renewable integration. In this context, real-time medium- and low-voltage grid monitoring and control under the new technologies will lead to significant improvement of grid efficiency and hosting capacity [8]. Moreover, local and timely management of distributed resources and self-consumption installations will be the key to ensure efficiency and stability at the distribution and the operation level [9]. Among the different solutions, PV self-consumption systems constitute a remarkable challenge for distributed generation contributing to future smart grids [10].

With regard to solar PV monitoring technologies, various review contributions can be found in the specific literature from different points of view [11,12,13]. Nevertheless, both environmental and technical factors can significantly affect the PV power generation—partial shading due to moving clouds, module temperature, humidity, the mounting angle, etc. Further explorations are then required to design an effective solar PV monitoring technology [14]. In addition, the use of monitoring systems in large-scale PV systems can be justified despite their high costs [15]. However, monitoring systems in medium and small PV installations may address additional and high costs unacceptable to most users. For this reason, data acquisition and monitoring solutions using low-cost platforms have been widely proposed in recent years [16]. The advance of technologies such as IoT and Industry 4.0 have allowed the implementation of new monitoring solutions as an alternative to traditional centralized systems. These solutions mainly aim to detect anomalies of PV solar plants and for maintenance purposes. Different approaches can be found in recent literature. Madeti et al. [11] provided a comprehensive review of PV monitoring systems, comparing most of PV monitoring evaluation techniques in terms of their relative performances. This contribution also examined sensors, controllers in data acquisition systems, data transmission methods and data storage and analysis. In [17], Sunarso et al. proposed a low-cost PV monitoring system based on an open-source Arduino platform and demonstrated its use for the assessment of PV monitoring and supervision potential. The system developed by Sunarso et al. can monitor solar irradiance, electric outputs and temperature of multiple solar panels. A similar approach can be found in [18], where the authors described a prototype for a portable data logger that integrates standalone instruments with open-source hardware technologies for monitoring PV systems. The proposed datalogger accomplished the accuracy requirements of the IEC standards for PV systems. In 2019, López-Vargas et al. [19] improved the design with a new version of this low-cost datalogger to overcome the shortcomings related to power consumption, the limited voltage range and the user interface for stand-alone PV systems located in remote areas deprived of telecommunications networks. Data were stored on an SD card to work autonomously. This work continued with a new design that included allowing 3G technology to monitor the PV systems remotely via web or via mobile application [20]. Regarding communications, Ansari et al. [14] reviewed various monitoring technologies for PV power plants focused on data processing modules and data transmission protocols. They showed key issues and limitations of this technology. In terms of transmission protocol, this work selected LoRa as a relevant data transmission solution, but it was not suitable for large payloads. Moreover, a continuous data packet sending process should be avoided due to some rules and constraints regarding the frequency band, which is in line with other recent contributions [21].

From the limitations of the techniques found in the specific literature, and considering the requirements of the application at hand, the authors previously proposed a solution using a Raspberry Pi that served as a comprehensive solar PV monitoring following the IEC-61724 requirements. The solution was used at the module level [22]. It was tested and assessed at real PV solar installations with remarkable results. The tests took some months in CETENMA SOLAR installations facilities. A 250 Wp module connected to the grid with a Soltec Solarfighter microinverter was monitored, as well as a 5 kWp ver-roof installation connected to the grid and located in the Universidad PolitÃľcnica de Cartagena (Spain) during some weeks. All variables were collected with standard equipment at both facilities. The results showed that the estimated error rate was lower than 2%. From this preliminary design, a low-cost LoRa-based solar PV monitoring system to communicate solar PV power plants located in remote locations was subsequently proposed in [23]. This topology used an Arduino Board with an RFM95W transceiver fabricated by HOPE RF configured as a LoRa TM modem available for the EU-868 MHz band [24]. The solution stored data in a packet size of 38 bytes with a transmission power of 14 dBm and spreading factor (SF) metric ranging from 10 to 12. An alternative system to be implemented in remote PV power plants for monitoring and dispatching electrical and meteorological data was then proposed and evaluated. This study demonstrated that the line of sight between source and destination and propagation issues had a clear influence on the suitable data reception process. As a main drawback, this solution only allowed a limited number of transmissions, due to the over-air time of works with high spreading factor value (SF11 and SF12) to ensure an accurate reception of data packets with a restricted duty cycle (1%). In addition, a LoRa sensitive analysis was conducted by the authors in [25] for PV short-time forecasting accuracy estimation.

By considering this preliminary background, as well as the lack of contributions focused on monitoring self-consumption PV solar installations and providing information to prosumers in terms of consumption/production profiles [26], the main contributions of this paper are described below:A novel monitoring solution for the operation and maintenance of self-consumption PV systems is proposed and assessed.
The solution is based on Internet of Things (IoT) applications by using Narrowband IoT (NB-IoT).The new design uses the Pycom IoT platform with NB-IoT as data transmission technology.

The Pycom development board FiPy and sensor shield Pysense have recently been proposed to monitor environmental variables such as humidity, temperature, altitude, pressure, or light [27].

The rest of the paper is structured as follows: Section 2 introduces NB-IoT technology; Section 3 discusses materials and methods; Section 4 describes the use case; Section 5 discusses both collected data and results; finally, conclusions are given in Section 6.

## 2. Communications: NB-IoT

Low-power, wide-area (LPWA) technologies are targeting most emerging markets and applications. LPWA is a recent generic term involving different technologies focused on enabling wide area communications, minimizing power consumption and cost [28]. LPWA has become a relevant growing space in IoT since in general LPWA solutions are perfectly suitable for such IoT applications that only need to transmit low amounts of data over a long range. Other researchers proposed a ZigBee network tested with WiFi for smart grid applications. However, these tools have limited capability for analyzing real scenarios and, according to Sultania et al. [29] ZigBee uses the same frequency band as WiFi, which can lead to potential radio interference. Consequently, most of the developed LPWA technologies have arisen in both unlicensed and licensed markets, such as SigFox, LTE-M, NB-IoT and long range (LoRa). Among them, Shina et al. [30] affirmed that NB-IoT and LoRa solutions are the two leading emergent technologies, despite the important technical differences in terms of network architecture, physical features and MAC protocol between them. Table 1 compares different network schemes and communication technologies.

Recently, different comparative studies are available in the specific literature evaluating LoRa and NB-IoT technologies [31]. Examples of new network architectures combining both technologies can be found in [32,33]. In terms of vulnerabilities, LoRaWAN and NB-IoT give sufficient security guarantees, but according to Coman et al. [34] both technologies need to be properly enforced. The NB-IoT technology was standardized in 2016 by the Third Generation Partnership Project (3GPP), considerably increasing the NB-IoT applications for wireless data communication purposes [35]. In addition, the presence of NB-IoT modules in the IoT device market share by 2030 will considerably increase too [36]. NB-IoT is compatible with LTE-M, with transmission speed of 150 kbps and a coverage range of 15 km. It is a licensed technology deployed in 79 countries, with an investment of $598 billion [37]. In this case, NB-IoT is selected by the authors for PV power plant monitoring and communication network evaluation. This proposal is in line with other recent network applications, such as Smart Water Grid (SWG) [38], water quality monitoring [39] or sustainable farming irrigation [40]. In all cases, NB-IoT was selected as a suitable candidate due to high scalability in comparison to other technologies, such as LoRaWAN and SigFox. Li et al. [41] demonstrated that NB-IoT satisfied both qualitative and quantitative requirements in terms of security, reliability and scalability. The relevant presence of IoT–infrastructure devices, the remarkable variety of applications and the use of various data processing solutions have recently led to the fact that traditional data center architectures do not allow the software configuration nor the physical scaling for tasks to be solved. Moreover, they are no longer able to provide the current required indicators focused on stability, controllability or productivity [42]. On the other hand, and regarding industrial applications, Ballerini et al. [43] concluded that NB-IoT offered the highest quality of service (QoS) while also ensuring data delivery, being a potential replacement to LoRaWAN when communication reliability was a required factor. In addition, NB-IoT was also proposed as an alternative solution for real-time demand response, able to monitor, control and connect electrical appliances [44].

In this paper, we propose an NB-IoT solution for PV power plant monitoring and communication purposes focused on self-consumption installations. Our approach is in line with recent contributions focused on PV installations [45,46,47]. We extend the use of IoT technology toward prosumers, providing an alternative solution for future distribution system scenarios in which a high renewable penetration would lead to higher peaks of generation as a consequence of potential reverse power flows at the medium voltage/low voltage (MV/LV) distribution transformers [48].

## 3. IoT Monitoring Solution

### 3.1. General Architecture

The proposed system uses a FiPy development board as the main hardware. FiPy is a low-cost ESP-32 development board produced by Pycom manufacturer, which includes WiFi, Bluetooth, LoRa, Sigfox and dual LTE-M (CAT M1 and NBIoT) technologies. This platform allows us to test different communication technologies excluding any hardware modification. Table 2 summarizes the most relevant technical features of these boards [49].

Pycom is programmed in MicroPython language [50]. For this purpose, it is necessary to use Pymakr, which is an Integrated Development Environment (IDE) plugin for popular code editors, such as Atom and Visual Studio Code. Pymakr thus makes the development of IoT edge devices running microPython easier [51]. Although it is possible to program FiPy via Pybytes, Visual Studio Code is an IDE widely used for this purpose. With regard to the code developed by the authors, Figure 1 shows the structure and main blocks of such code schematically. Firstly, the sensors are configured and set, the FiPy microcontroller being able to collect data from such sensors. Secondly, a reading process of data from the sensors is carried out, and subsequently, the corresponding message is sent via LTE including the gathered data. It is then necessary to verify if the device is connected to the Pybytes platform. Indeed, eventual disconnections were detected during the testing, and a reconnection process was included in the following terms: Once the connection is verified, the message can be sent to the platform via LTE. This process is repeated with one-minute sample time during the total of sunlight daily hours. FiPy is set on deep-sleep under low sunlight conditions, in line with other contributions [52].

Pybytes is a free cloud-based device management platform for all Pycom development boards and modules. It provides a mobile app, allowing us to manage devices directly from a smartphone. It is possible to change the priority of the used networks: WiFi, Bluetooth and LPWAN networks. Pybytes also provides an easy to use dashboard to quickly create an application that sends data to the platform and allows us to choose among different types of data visualization. That dashboard updates automatically, and subsequently, the user can visualize current status of the devices and received data in real time. It is also possible to integrate Pybytes with a Cloud provider: AWS, Microsoft Azure, Webhooks and Google Cloud. A recent infrastructure based on Pycom development board FiPy and sensor shield Pysense to collect and send data to the remote cloud over Wi-Fi and Long Range (LoRa) protocols is also described in [27] for remote monitoring conditions. In this case, to provide greater reliability and synchronicity to the data, a DS1302 RTC was incorporated. Thanks to this module, the sending of packets to Pybytes was controlled every minute as were the deep-sleep mode periods for suitable energy saving. This module provided us with a temporary time stamp, helping us when drawing conclusions from the measured data. In addition, and due to the constant disconnections of the Pycom LTE network, different solutions were tested by the authors. The connection to the network was configured by following the official documentation. The script was set to send packets every minute over NB-IoT. In parallel, the firmware of the FiPy microcontroller used was developed to perform a ping to the Pybytes platform and thus verify correct communication. If that ping was not answered due to any connection drop, then it would wait 10 min for the next ping. If this second ping was not answered either, then the watchdog timer was triggered, and the system rebooted to re-establish communication. Consequently, under a communication failure event, 10 min of information would be lost as the microcontroller remains on hold. This operation cannot be modified in the program, but it is a matter of firmware configuration.

### 3.2. Hardware

With regard to hardware components, the selected sensors as well as the calibration process and the prototype encapsulation are now described. In fact, the selected sensors provide an analog output necessary to calibrate them, in our case by using the Transmille 3000A Series calibrator. The sensors are in charge of monitoring PV installations under the IEC-61724 requirements. The IEC-61724 standards were previously proposed to analyze PV system performance under a variety of climates [53]. In our solution, PV electrical data and weather parameters are gathered to estimate PV operating conditions and exchange meteorological and electrical data. More specifically, the ACS758 current sensor [54] and the YHDC HV25 voltage sensor were selected.

#### 3.2.1. DC-Output Current: Acs758 Hall-Based Effect Sensor

A breakout board for the ACS758 Hall effect linear current sensor was selected for measuring the DC current of the PV module string. The thickness of the copper conductor of this sensor allows the device to survive under high current conditions. The ACS758 provides an analog voltage output signal that varies linearly with the current value. The output voltage of the sensor varies significantly depending on the input current value. Therefore, different currents were applied within the available measurement interval, ranging from −50 A to 50 A DC. In this case, we used a calibration current range from 0 A to 20 A, in steps of 1 A. Subsequently, the sensor output voltage was measured through the FiPy microcontroller, see Figure 2. With this aim, a script was defined to convert the measured value into an estimated current value. The sensor sensitivity provided by the manufacturer was 40 mV/A, for a channel up to 5 V. In our case, the microcontroller channel input was 3.3 V, and subsequently the sensitivity to be used was 26.4 mV/A. The offset voltage was 1.65 V for 0 A, corresponding to 50% of the FiPy microcontroller analogical offset range. The relative error was estimated by comparing the real input current to the measured current values, with a maximum error of 4%. Different current sensors were calibrated and compared to evaluate their performance and accuracy within the expected current range. Figure 3 compares two current sensor calibration processes. An additional source of potential errors in the current sensor output is the sensitivity of the FiPy analog input channel, which is 20 mV/A.

#### 3.2.2. DC-Output Voltage: Yhdc Hv25 Sensor

Regarding the DC voltage measurement, the low-cost YHDC HV25 sensor was selected. It allows us to measure voltages up to 1000 V, scaling them to 5 V. Since the analog inputs were used according to the Pycom FiPy microcontroller—working at 3.3 V, it was necessary to include a voltage divider circuit between the sensor output and the analog–pin input. This voltage divider consists of a 56 kΩ resistor and two 56 kΩ resistors in series. In this way, an output of 3.3 V is obtained for an input of 1000 V, with a sensitivity of 3.3 mV/V. With the aim of calibrating the sensor, a set of measurements were carried out from from 0 V to 1000 V, in steps of 50 V. For this purpose, a symmetrical voltage supply of ±12 V and a variable voltage output signal were used together with the 3041A precision multi-product calibrator, see Figure 4. A set of input voltage measurements were compared to the real DC voltage to obtain a calibration curve for this voltage sensor. In a similar way to Section 3.2.1, an additional set of measurements were also collected to estimate the relative error under different boards. By considering that each solar panel had an open-circuit voltage ($V_{oc}$) of 50 V, the determined relative error was lower than 3%.

Due to the high error obtained in the low range of measurement—lower than 50 V—the option of recalculating the voltage divider was then proposed by the authors. The target was to equate the maximum input of 3.3 V to 800 V to test if the sensor precision could be improved below 50 V. After a series of testing measurements, the option was finally discarded since they did not significantly improve the results, and there was a risk of obtaining a voltage higher than 800 V that could damage the equipment. It was observed that the selected sensor did not provide a suitable voltage output for inputs lower than 50 V. Therefore, a voltage divider was included in the laboratory tests to allow collecting voltage data below such a threshold instead of using the YHDC HV25 sensor. This voltage divider was formed by two resistances: 10 kΩ and 1 kΩ, respectively. The output signal was then collected from the 1 kΩ voltage value. Given the high variability of the error for each YHDC HV25 plate, the option of calibrating each board individually, with its own calibration line, was selected. With this aim, Figure 5 compares the calibration process for three different voltage sensor boards. In addition, Table 3 shows the estimated relative errors for these three different voltage sensor boards. As shown in these results, the proposed board has minor accuracy at low voltage levels but high accuracy over the expected operating voltage ranges in medium and small PV installations. Therefore, it is accepted for our proposals.

#### 3.2.3. Solar PV Module Temperature: Ds18b20 Digital

The DS18B20 is a digital thermometer providing from 9-bit to 12-bit Celsius temperature measurements up to 125 °C [55]. There is also a watertight package that protects the sensor and allows it to be submerged in a liquid without damage available. Since it is a digital sensor, the read signal does not degrade due to wiring distance. It can work in one-wire mode with an accuracy of ±0.5 °C and 12-bit resolution. Various sensors can also be used on the same pin, as they can be internally programmed with a unique 64-bit ID to identify and differentiate them. The operating range is 3 to 5 V, being able to be used in virtually any system by the use of microcontrollers. Further information of temperature sensor analysis for PV solar module temperature measurement can be found in [56]. The DS18X20 class from the onewire.py library [57] was also included to properly use this sensor with the FiPy. The code can be found in the Appendix A.

#### 3.2.4. Environmental Variables: Sht3x Sensor

The SHT31 is a combined humidity and temperature sensor specially designed for outdoor applications. Recent uses can be found in the building of monitoring applications [58,59]. Its performance characteristics allow the sensor for outdoor applications. It can work on 3.3 V and 5 V systems with very low power demand, offering fast and accurate measurement via I2C bus. The I2C class found in the micropython machine library was included to use this sensor, in addition to the SHT3X.py library available in [60]. The developed script can be found in the Appendix B. The irradiance is measured by a reference solar panel (W/m${}^{2}$). This small PV module of 300 mW is directly connected with a shunt resistor for sampling the voltage generated by the sunlight—68 × 37 mm, mono-crystalline cells; 19% efficiency, 5 V, 60 mA. The analog signal is directly dependent on the irradiance. Further information can be found in [61]. It is calibrated using a pyranometer located close to the plate and in the same position, being then possible to relate the current generated by the reference plate and the available solar irradiance, see Figure 6.

### 3.3. Economic Evaluation: Cost-Effectiveness

Finally, and in terms of cost-effectiveness, the proposed monitoring system is in line with other contributions discussed in Section 1 and Section 2. The proposed system is flexible to be configured in different locations and PV installations. With the aim of offering a low-cost system, the different hardware components are selected from open-source projects with a high cost effectiveness threshold. Table 4 gives the monitoring node cost, which is lower than other commercial solutions but higher that previous developments carried out by the authors [22,23]. Nevertheless, the global cost could be reduced by using other open-source platforms—such as Arduino—and depending on the number of nodes to be produced and/or purchased.

## 4. Use Case Description

With the aim of evaluating the capability of the proposed solution in terms of measuring, collecting and transmitting data, a preliminary prototype with na FiPy was developed by using a protoboard. Some additional modifications and requirements were detected during this evaluation process. As an example, the inclusion of a potential divider to measure the voltage sensor output, as well as the necessity of increasing an additional input voltage for this sensor were considered, see Figure 7. After the preliminary assembly was tested, a first printed circuit board prototype was designed and milled to minimize the number of wires required by the system by using a computer numerical control (CNC) milling machine. In this case, the printed circuit board (PCB) design was carried out by using Fritzing [62]. It is an electronic design software including libraries for different elements. Flatcam was also used to create the GCode, finally used on the CNC machine that milled the PCB. Different versions were proposed and tested to improve different aspects of the design, mainly reducing sizes and elements required for the solution. Figure 8 shows the connection diagram schematically.

Electronic components need to be protected from moisture, direct sun radiation, dust and other external effects that could affect their normal operation. Moreover, to avoid any interference in control signals and to isolate those components under high voltage or intensity values, two different external boxes were designed using Onshape. It is a free-to-use CAD platform in which assemblies can be created by using different items to make a more complex design. Different recent applications can be found in the specific literature [63].

With the aim of providing power to the node, a PV solar module (6.5 W rate power) with a 2000 mAh Li-Ion Polymer battery was also included. The battery was selected to ensure system operation at night or during periods when solar radiation is not enough to provide the required energy under operating conditions. To ensure a relevant autonomy of the system, the micro-controller goes into deep sleep at night and under low radiation conditions. Figure 9 shows the prototype battery and power supply scheme.

## 5. Results

The proposed solution—both including hardware components and sensors—was tested in a laboratory environment, aiming to assess its performance under different conditions. The solution was connected and assembled under outdoor conditions in a PV self-consumption installation to evaluate the solution feasibility to be implemented in a variety of real situations and conditions. The proposed system was initially tested at the solar laboratory of CETENMA, located in the Industrial Park of Cartagena (Spain). Such a facility includes measurement equipment to check the performance of PV power plants and modules. For testing purposes, a single 250 Wp monocrystalline PV module connected to an SF 250 W Soltec SolarFighter microinverter was used—V${}_{MPP}$ = 28.5 V, I${}_{MPP}$ = 8.8 A, V${}_{oc}$ = 34.6 V, I${}_{sc}$ = 9.4 A. Figure 10 shows some examples of field test campaigns and Figure 11 shows results corresponding to this calibration process.

From the power supply scheme shown in Figure 9, Figure 12 depicts voltage evolution and current demanded (positive) or supplied (negative) by the battery. The Pycom module consumption is around 200 mA over WiFi and 250 mA when the LTE communication protocol is selected. During the information sending process, up to 350 mA can be achieved. Figure 13 shows a detail of Pycom and battery power demand comparison. Regarding the received data through NB-Iot communications versus inverter data, voltage and current have enough precision, 2.45% and 3.9%, respectively, to evaluate the operation of the plant with security. A total of six packages were sent with one-minute sample time: 8 bytes/package for five packages and 10 bytes/package for the additional package. In terms of electrical collected data, Figure 14 and Figure 15 show both voltage and current data respectively. Even though the selected reference cells achieved suitable results during the calibration process, the received data show divergences during some periods of the day. These divergences are due to both material and design of the enclosure of the reference cell that, after some time, does not ensure that it remains in the plane of installation with modules of the plant, reducing the received radiation with low values of solar angles, see Figure 16. A new design for this enclosure is currently in progress. In terms of temperature gradients, Figure 17 compares PV module temperature to ambient temperature These data give additional information regarding PV operating temperature, which can be useful to estimate other parameters, such as the performance ratio. In addition, it is also possible to analyze the effect of irradiation and ambient temperature on PV system performance, as was suggested by other authors [64,65].

Regarding communication quality of the proposed solution, additional metrics used in previous works of the authors [22] are also discussed because no data of works with similar systems have been found to compare the results. These metrics are Packet delivery ratio, defined as the ratio between the packets successfully received and the total data packets sent by the end nodes and time intervals between different data packets (inter-arrival time) that is determined by the time interval value corresponding to each packet received. In the previous work that used Lorawan communications [22], the payload had a length of 38 bytes, almost equal size of this work, where the size of the sent string was 37 bytes, 4 × 5 floating bytes (measurements of each sensor) and 17 bytes of temporary fingerprint. With these similar conditions, our study reveals that the proposed system provides a reliable connectivity with a packet delivery error around 4.6% and stable time interval between packets of 60 s. These results validate the feasibility and reliability of our proposal improving the results obtained in previous works and overcoming limitations related to missing data packets and frequency of received data. Figure 18 shows these results.

## 6. Conclusions

The proliferation of PV installations either for self-consumption or with the aim of alleviating dependency on the grid is moving the energy system to more decentralized power generation. This new scenario allows users (who now are prosumers) to generate and/or consume their own energy. The lack of information about these PV systems emerge as a relevant challenge for the management of micro- and medium-size grids, as the generation is more distributed. This paper describes a monitoring solution suitable to be applied for self-consumption or any other micro-generation installation, covering the installations of the so-called ’prosumers’ and aiming to providing them with a tool that informs them about their local self-consumption. The proposed system allows us to monitor both electrical and environmental variables. A prototype was calibrated and successfully evaluated in a real PV self-consumption installation. Both current and voltage sensors were calibrated, determining 3% and 4% relative error, respectively, under laboratory conditions. The system includes a 6.5 W PV solar panel and a battery for energy requirements. Charge and discharge battery cycles were also monitored and included in the paper. Regarding the received data through NB-IoT communications versus inverter data, voltage and current have enough precision, 2.45% and 3.9%, respectively, to evaluate the operation of the plant with security. In addition, the proposed system provides a reliable connectivity with a packet delivery error around 4.6% and stable time interval between packets of 60 s. The size of the string was 37 bytes: 17 bytes of temporary fingerprint, 4 × 5 floating bytes. This solution is also able to be implemented in large PV power plants, as well as other alternative renewable installations.

## Figures and Tables

**Figure 1 sensors-22-04966-f001:**
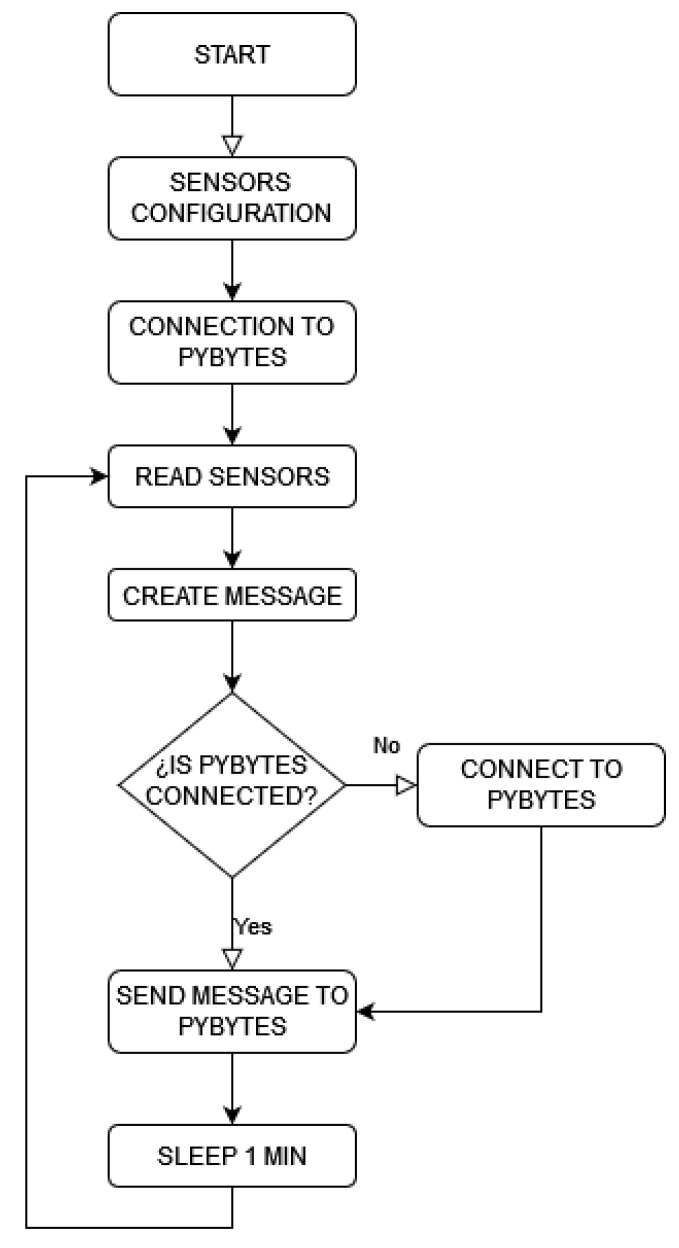
Flow diagram for the microcontroller.

**Figure 2 sensors-22-04966-f002:**
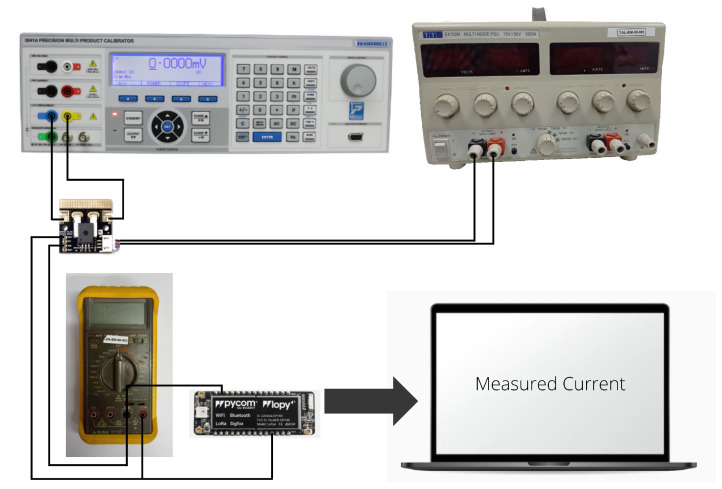
Current sensor calibration assembly: Transmille 3000A Series calibrator, ammeter, regulated power supply and current sensor.

**Figure 3 sensors-22-04966-f003:**
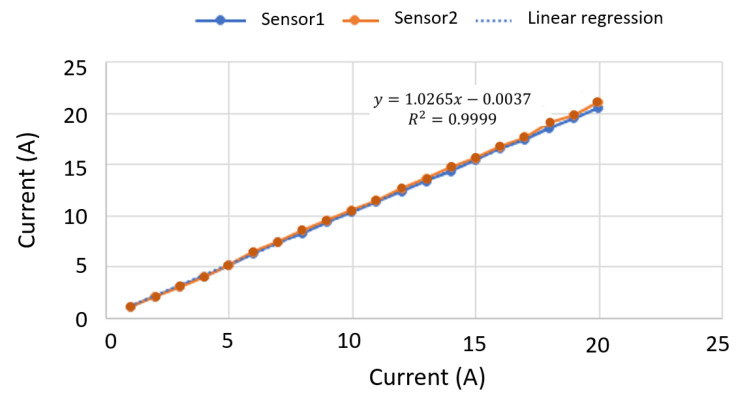
Current sensor calibration process: comparison of input current vs measured current.

**Figure 4 sensors-22-04966-f004:**
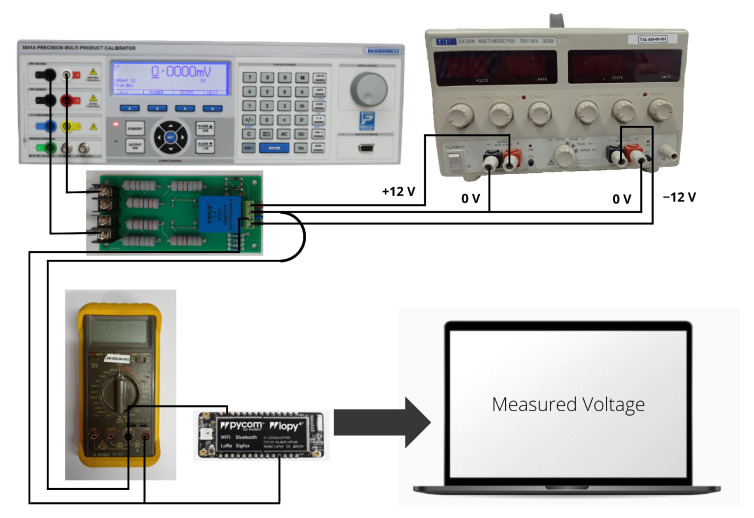
Voltage sensor calibration assembly. Transmille 3000A Series calibrator and voltage sensor.

**Figure 5 sensors-22-04966-f005:**
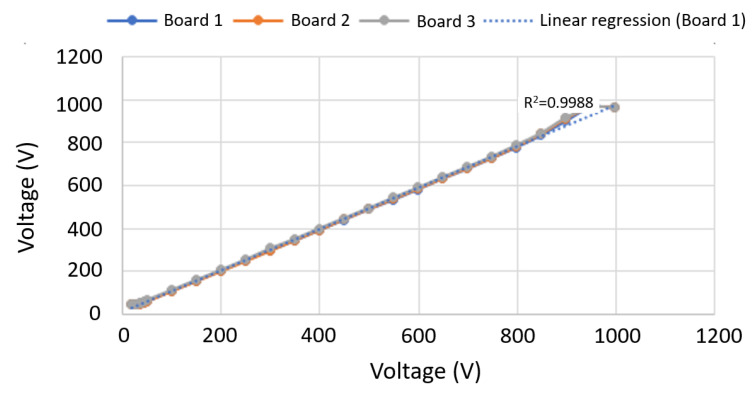
Voltage sensor calibration process: comparison of input voltage vs measured voltage.

**Figure 6 sensors-22-04966-f006:**
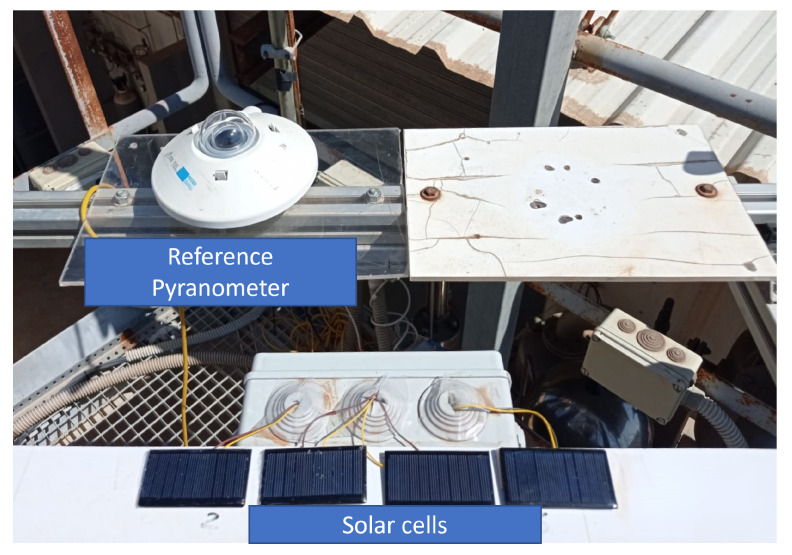
Solar panel calibration:lLaboratory environment.

**Figure 7 sensors-22-04966-f007:**
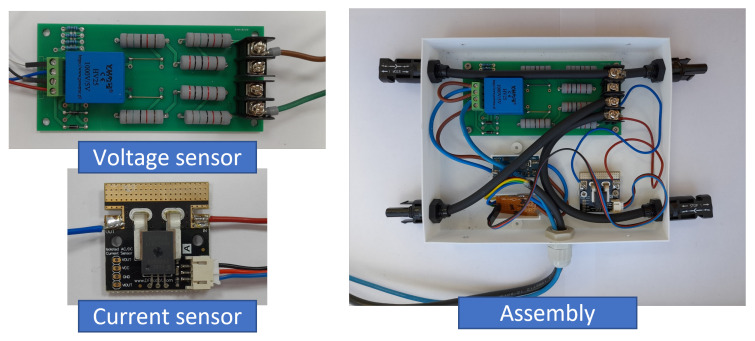
Prototype voltage and current sensors.

**Figure 8 sensors-22-04966-f008:**
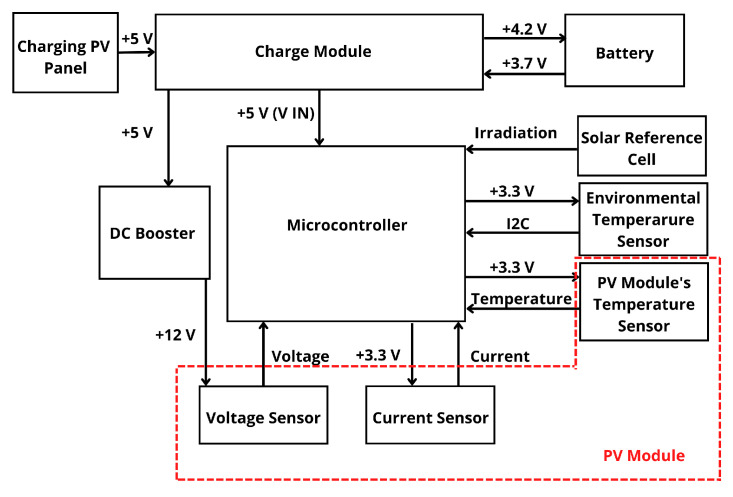
General scheme of the system electronics.

**Figure 9 sensors-22-04966-f009:**
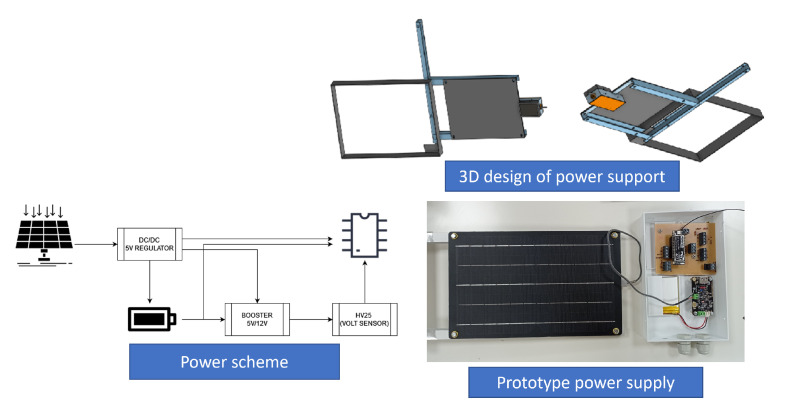
Power supply scheme of the system and prototype example.

**Figure 10 sensors-22-04966-f010:**
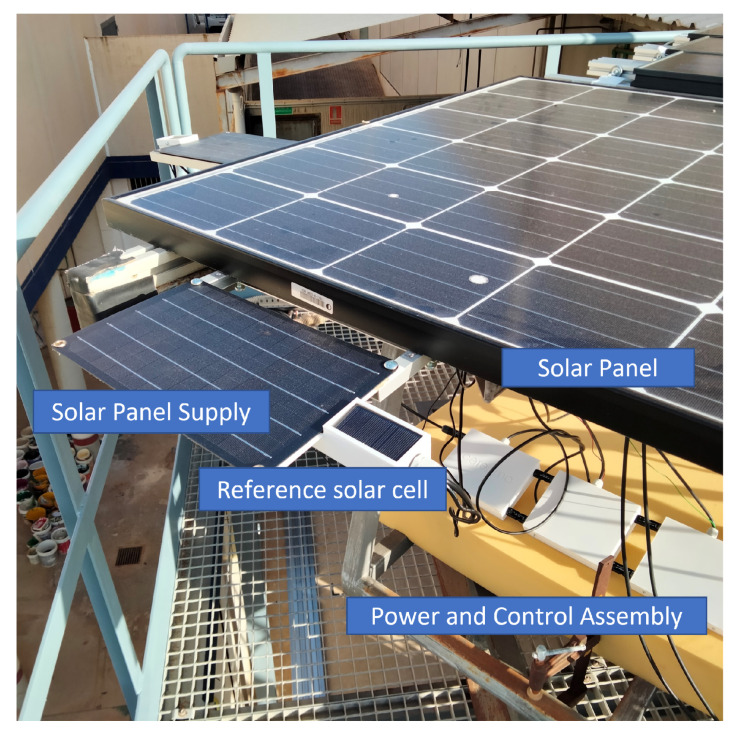
Case study. Example of field test campaign. Charging solar panel and reference cell assembled to the monitored PV module panel.

**Figure 11 sensors-22-04966-f011:**
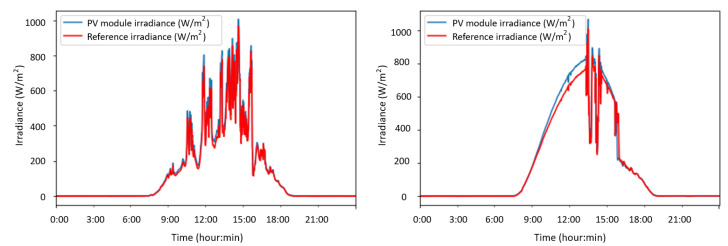
Calibration of the reference cell: comparison of data for two days.

**Figure 12 sensors-22-04966-f012:**
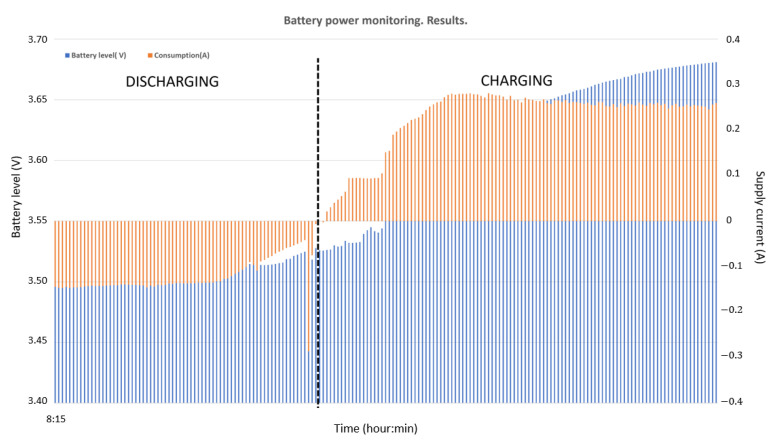
Example of battery charge and discharge time periods.

**Figure 13 sensors-22-04966-f013:**
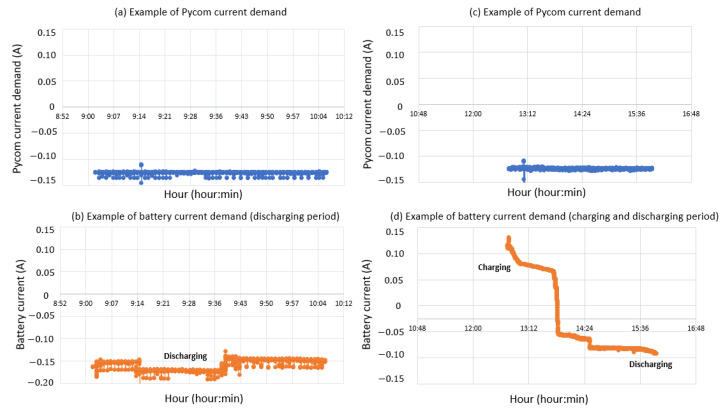
Examples of Pycom demand consumption and battery power monitoring: (**a**,**b**) discharging battery period; (**c**,**d**) charging and discharging battery period.

**Figure 14 sensors-22-04966-f014:**
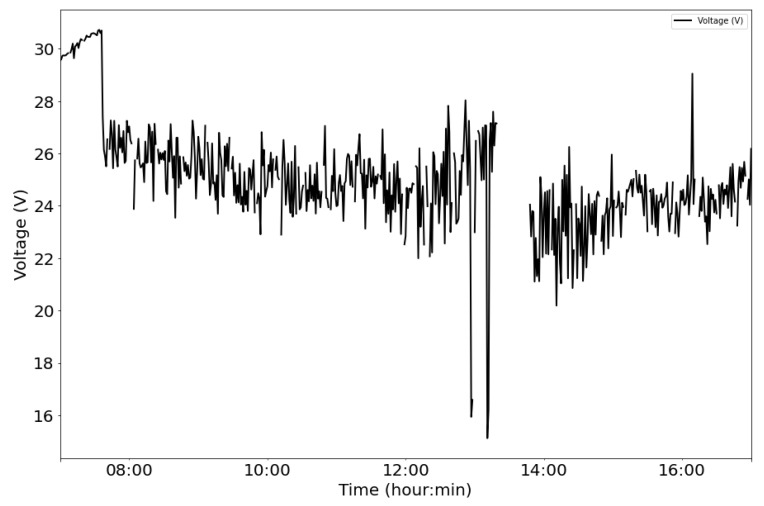
Example of collected voltage data: PV module monitored.

**Figure 15 sensors-22-04966-f015:**
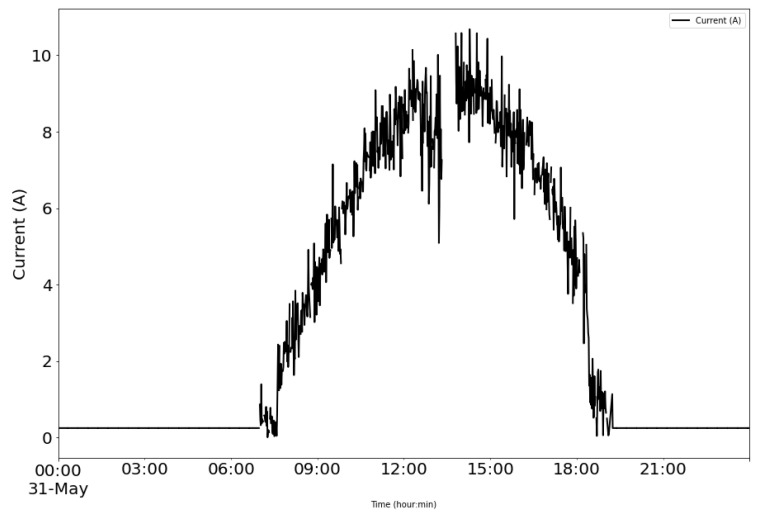
Example of collected current data: PV module monitored.

**Figure 16 sensors-22-04966-f016:**
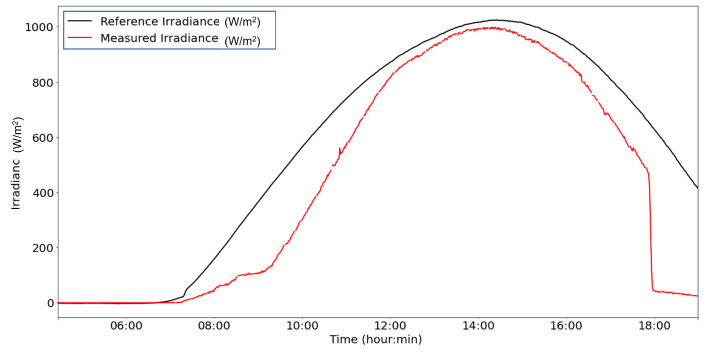
Example of collected data received in-plane irradiance.

**Figure 17 sensors-22-04966-f017:**
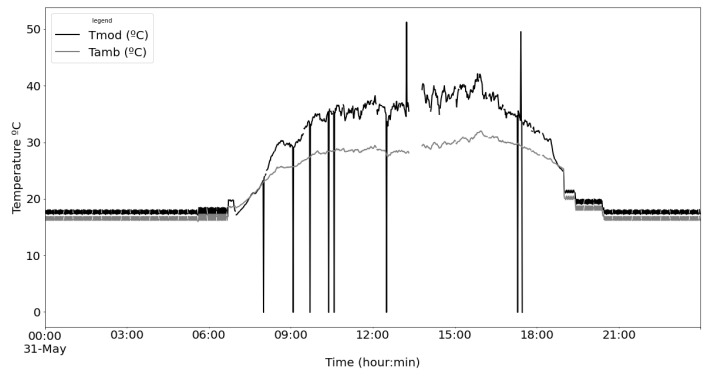
Comparison of PV module temperature vs. ambient temperature.

**Figure 18 sensors-22-04966-f018:**
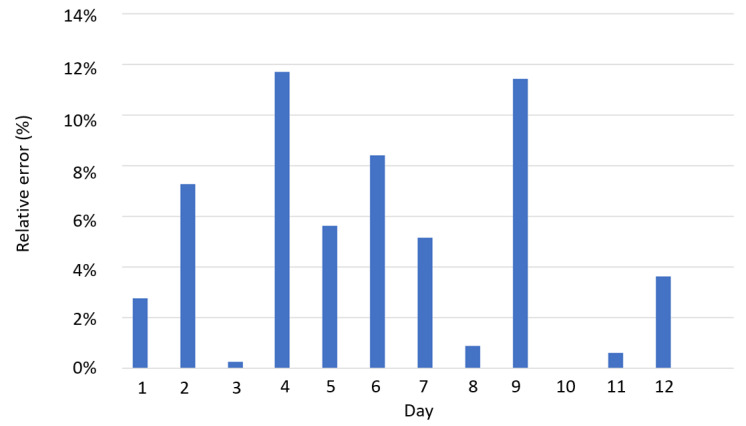
Communication relative errors: Average daily values.

**Table 1 sensors-22-04966-t001:** Network schemes and communication technologies properties.

Network Schemes	Technologies	Data Transfer Rate	Typical Coverage Range	Latency
HAN	Ethernet, PLC, Zigbee, WLAN, Z-Wave	10–100 kbps	up to 500 m	10 ms–1 s
BAN/IAN	Ethernet, PLC, Wimax, WLAN	100 kbps–1 Mbps	up to 1 km	10 ms–2 s
NAN	Ethernet, PLC, DSL, Fiber–Optics, WiMax, NB–IoT, LoRa	100 kbps–10 Mbps	0.1–10 km	10–50 ms
WAN	PLC, Ethernet, Fiber–Optics, LoRa, WiMax	10 Mbps–1 Gbps	10–100 km	10 μs–20 ms

**Table 2 sensors-22-04966-t002:** FiPy specifications.

CPU
- Xtensa*©* dual–core 32–bit LX6 microprocessor(s), up to 600 DMIPS
- Hardware floating point acceleration
- Python multi–threading
- An extra ULP–coprocessor that can monitor GPIOs, the ADC channels and control most of
the internal peripherals during deep–sleep mode while only consuming 25 μA.
**Networks**
- WiFi (1 km range)
- BLE
- Cellular LTE-CAT M1/NB1 (Total world–wide support)
- LoRa
- Sigfox
**Memory**
- RAM: 520 kB + 4 MB
- External flash: 8 MB
- GPIO: Up to 22
- Hardware floating point acceleration
- Python multi–threading
**Interfaces**
- 2 ×UART, 2 × SPI, I2C, micro SD card
- Analog channels: 8 (12–bit ADC), 2 (8–bit DAC)
- Timers: 2 of 64 bit with PWM with up to 16 channels
- DMA on all peripherals
- GPIO: up to 22
**Hash/Encryption**
- SHA, MD5, DES, AES
**RTC**
- Running at 32 kHz
**Range**
- Node range: up to 50 km
**Power**
- Voltage Input: 3.3 V–5.5 V
- 3v3 output capable of sourcing up to 400 mA
**Size**
- 55 mm × 20 mm × 3.5 mm (excluding headers)

**Table 3 sensors-22-04966-t003:** Relative Error YHDC HV25.

Range (V)	Board 1	Board 2	Board 3
50–950	2.93%	2.66%	2.83%
150–1000	2.51%	2.57%	1.49%

**Table 4 sensors-22-04966-t004:** PV monitoring node cost.

Description	Unit Price (Euro)
Mainboard: FiPy	59.4
Sensor: Irradiance	2.45
Sensor: PV DS1820 temperature sensor	1.5
Sensor: DHT temperature and humidity sensor	0.18
Sensor: Voltage sensors (DC)	18.4
Sensor: Current (DC)	17.5
Supply: PV module	9.95
Supply: Battery	6.9
Supply: DC/DC converter	7.9
Supply: DC Booster (24 V)	0.18
Outdoor enclosure and wiring	12
Total cost	136.36

## Data Availability

The data that support the findings of this study are available from the corresponding author, A.M.-G., upon reasonable request.

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
