# Peer review of "Democratization of PV Micro-Generation System Monitoring Based on Narrowband-IoT"

_sensors, 2022, doi:10.3390/s22134966_

Round 1

Reviewer 1 Report

This manuscript describes the design and implementation details of an IoT-integrated sensor for possible use in photovoltaic panel monitoring at the prosumer level. The presented work fits well the scope of the sensors Journal. However, it needs extra effort to be improved and presented properly. I have the following concerns that must be carefully considered by the authors.

1.    The abstract is too long with many abbreviated terms. Therefore, any abbreviated term should be fully phrased before using it anywhere in the manuscript. There are many examples of that such as “DSOs, TSOs, IEC, LTE, LTE-M, MAC, ……. etc.”.

2.      Some keywords are not suitably selected such as “open–source” and “low–cost solutions”, as they are too generic.

3.    Many typos and grammatical errors are there in the manuscript. Many examples can be found such as

“…. and more specifically the Activity 4: Operation….”,

“….. factors can affect considerable the PV power generation…..”,

“ ….. and high costs not acceptable for most of users.”,

“….were stored in a SDCARD to …”,

“…..remotely via web or via mobile application” ,

“ (Internet of the Things),

“…wide-asrea…”,

“…there is not sunlight to..”…………etc.

4.    Many long and unclear sentences need to be reformulated such as that sentence started from lines number 175 to 179. Generally, such long sentences should be avoided.

5.    The authors’ contribution is merged with the literature review and not clearly identified. Therefore, the authors must clarify their contributions in specific points. 

6.   In Table 2, the authors listed many specifications of the used development board, however, they didn’t mention anywhere in the text how these specifications assisted or affected the results of their system performance.

7.   In sections 2 and 3, the authors always return to the literature and didn’t focus on their own work. They referred to many topics and terminologies, however, nothing can be found in the results section about these raised issues. Examples of that are LoRaWAN, SF, SigFox, QoS, MAC protocol, NB-IoT, …. etc.

8.   In lines 219 and 220, it is mentioned that 10-minutes are required to reconnect to the network. This period is considered to be long enough so the proposed sensor solution misses many events during these reconnection attempts, for example, sudden clouds shading.

9.    All of the presented figure captions are not informative. These captions must be rewritten to clearly describe the figure information rather than isolated words or broken sentences.

10. In Figure 2, the measurement unit of the multimeter is hidden. It must be clarified whether it is in volts or millivolts. Along with the experiment setup, it is helpful to add a schematic diagram to illustrate the calibration connection. The displayed calibration numbers (15 A and 2.773) should be discussed in the relevant section.

11.In line 251, it is mentioned that the current error is 4%. According to the datasheet of the ACS758-050B current sensor, the output error is much lower than that. Therefore, the source of this additional error should be justified. Please check if an output filter for noise management is essential.

12.Figure 3, and the similar ones, should be labeled as 3(a) and (b) and clarified in the figure caption.

13.In Table 3 and its corresponding text, it is not clear why the authors need three boards for the voltage sensors. In addition, there is no clear solution mentioned to solve the problem of inaccurate calibration results for the low-voltage range (< 50 V).

14.   The authors presented voltage sensor calibration up to 1000 V. However, they tested this sensor only on voltage less than 28 V. Additionally, there is no discussion on the results anywhere in the manuscript.

15.In figure 4, an input voltage connection is seen but the output calibrated voltage is not shown.

16.In Figure 5, it seems the sensor suffers from saturation in the low voltage range. There is no comment on these results.

17. In Figure 6, and similar figures, any related component should be labeled within the displayed pictures.

18. In Figure 8, the connection diagram must be re-organized to clarify VCC, GND, and signal tracks should be labeled. Additionally, I urge the authors to introduce a block diagram (in the architecture section) comprising all system components and their relation to each other.

19.   In my opinion, some displayed figures are redundant or unnecessary as they don’t have a strong correlation with the discussed topics. Examples of that are figure 9 (CNC machine), figure 10 (box design), and figure 11 (panel assembly). Instead, the authors should use the space to describe well their system and present a helpful dissection of the results.

20.   There is no information about the covered range of the radio frequency signal or connection of an external antenna to the development board.

21. Again, there is no adequate information on the PV panels used to implement or test the system. For example, open-circuit voltage, short-circuit current, thermal coefficients, current-voltage characteristics, …... etc.

22.   What is the purpose of Figures 15 and 16? Charging and discharging a small battery is not evidence of the proper system operation. How about the performance of that battery in absence of irradiance for two days?

23.   The authors discarded important results such as evaluation of packet-error rates, latency, quality of service, IoT platform protocols, system reliability, security … etc.

24.   What are the power consumptions of the proposed system in case of normal operation and worst-case scenario?

25.   The authors tested their monitoring system on a single 250 W PV panel, how about the medium size PV system mentioned several times in the context of their manuscript.

26.   Figure 17 should be displayed on the IoT platform dashboard for at least 12 hours.

27.   In figure 18, what is the difference between the two sub-figures? There is no adequate information on that.

Author Response

Please, find attached our response. Thank you.

Reviewer 2 Report

The paper is quite interesting. It considers questions of a monitoring solution that is suitable to be applied for self-consumption or any other micro-generation installation. The authors proposed a system at the module level, which was provided in detail, and extended information regarding low–power wireless area network application to the PV power plant monitoring issue, being one of the most notable innovations of this work.

The paper is technically sound. The paper contributes to the body of knowledge. It presents new practical results. The provided references are applicable and sufficient.

There are some comments:

1) Figure 8 and Figure 12 must be enlarged; they should be contrasting and readable.

2) The labels on the graphical axes in Figure 3, Figure 5, and Figures 15-18 are poorly visible.

3) In Section 5 Results the authors should add a table with data to compare the obtained results of the proposed system with parameters of already known systems for monitoring micro-generation installation based on NB–IoT. The authors should show the advantage of the system they proposed by the comparative analysis of the data from this table.

4) Numerical data of the proposed system must be presented in the Abstract and Conclusions.

It is a good study. I will recommend this paper for acceptance in the Sensors journal after minor correction.

Author Response

(The authors gave the same response as above.)

Round 2

Reviewer 1 Report

In the modified version of their manuscript, the authors presented additional information and made extra effort to improve and clarify their work. However, some untreated points are still need some attention as described below.

1.    Number of the keywords should be increased.

2.    Some typos errors are still there in the manuscript. Examples on that are:

     The word “based” was repeated so many times (40 times).

     In line 140 “………The solution is based based on Internet……..”

3.    The manuscript must be reorganized so that the related tables and figures appear after (not before) the explaining text. Moreover, all of the figures should be placed before the conclusion section.

4.    Some figures are not clear enough and need to be prepared in high resolution. Examples on that are figures 2,4, and 8. In figure 8, it is not clear whether the voltages sign is positive or negative. I encourage the authors to improve resolutions of all figures.

5.    Again, some of the presented figure captions are not informative. These captions must be rewritten to clearly describe the figure information rather than the isolated words like “… Result.”. Examples are captions of figure 3, 5, 6, 10, 11, 12, 13, 14,15, 17, and 18.

6.    Again, in figure 9 and 13, should be labeled as (a), (b), (c) or (d). Each sub-figure should be clarified in the figure caption an explained in the reverent text.

7.    In Table 3, the boards testing exhibit different errors for different ranges. However, figure 5 presents the same results for all of the boards. This figure should be treated to clarify the difference in board testing accuracy.

Author Response

(The authors gave the same response as above.)
